# Understanding genetic risk factors for common side effects of antidepressant medications

Adrian I. Campos [1,2 ✉], Aoibhe Mulcahy[1,3], Jackson G. Thorp[1,2], Naomi R. Wray [4,5], Enda M. Byrne[4,6], Penelope A. Lind[1], Sarah E. Medland [1], Nicholas G. Martin [1], Ian B. Hickie[7] & Miguel E. Rentería [1,2 ✉]

## Abstract

**Background** Major depression is one of the most disabling health conditions internationally. In recent years, new generation antidepressant medicines have become very widely prescribed. While these medicines are efficacious, side effects are common and frequently result in discontinuation of treatment. Compared with specific pharmacological properties of the different medications, the relevance of individual vulnerability is understudied.

**Methods** We used data from the Australian Genetics of Depression Study to gain insights into the aetiology and genetic risk factors to antidepressant side effects. To this end, we employed structural equation modelling, polygenic risk scoring and regressions.

**Results** Here we show that participants reporting a specific side effect for one antidepressant are more likely to report the same side effect for other antidepressants, suggesting the presence of shared individual or pharmacological factors. Polygenic risk scores (PRS) for depression associated with side effects that overlapped with depressive symptoms, including suicidality and anxiety. Body Mass Index PRS are strongly associated with weight gain from all medications. PRS for headaches are associated with headaches from sertraline. Insomnia PRS show some evidence of predicting insomnia from amitriptyline and escitalopram.

**Conclusions** Our results suggest a set of common factors underlying the risk for antidepressant side effects. These factors seem to be partly explained by genetic liability related to depression severity and the nature of the side effect. Future studies on the genetic aetiology of side effects will enable insights into their underlying mechanisms and the possibility of risk stratification and prophylaxis strategies.

### Plain language summary

Antidepressants are commonly prescribed medications, but adverse side effects are cause for treatment discontinuation. We analysed data from a large group of adults who have taken antidepressants to understand why some people experience specific side effects. Our results suggest that a person's genetic characteristics play a role. For example, participants genetically predisposed to a higher body mass index were more likely to report weight gain from antidepressants. These results open up the possibility of predicting adverse side effects as we increase our knowledge on the genetics of related complex traits. Future studies can focus on performing large-scale genetic studies of antidepressant side effects to gain further insights into the mechanisms underlying antidepressant side effects and to identify genetic markers of side effects that could be used in the clinic.

[1] Department of Genetics and Computational Biology, QIMR Berghofer Medical Research Institute, Brisbane, QLD, Australia. [2] Faculty of Medicine, The University of Queensland, Brisbane, QLD, Australia. [3] School of Biomedical Sciences, Faculty of Health, Queensland University of Technology, Brisbane, QLD, Australia. [4] Institute for Molecular Bioscience, The University of Queensland, Brisbane, QLD, Australia. [5] Queensland Brain Institute, The University of Queensland, Brisbane, QLD, Australia. [6] Child Health Research Centre, The University of Queensland, Brisbane, QLD, Australia. [7] Brain and Mind Centre, University of Sydney, Camperdown, NSW, Australia. ✉email: adrian.campos@qimrberghofer.edu.au; miguel.renteria@qimrberghofer.edu.au

The World Health Organisation predicts that depression will become the leading cause of disability globally by 2030[1]. The symptomatology, longitudinal course, response to treatment and functional impact of depressive disorders are highly variable. Antidepressant medicines are widely prescribed across the spectrum of depression severity and subtypes, alone or in combination with psychological therapies.

Effective pharmacotherapies for depression were first developed in the 1960s, following the identification of antipsychotic therapies. A clear focus was on the regulation of brain monoamine systems (dopamine, serotonin and noradrenaline). These agents, including tricyclic antidepressants (TCAs) and Monoamine Oxidase Inhibitors (MAOIs), were limited in the extent of their use by considerable side effect burdens and potential toxicity. From the 1980s onwards, further pharmacological developments have been dominated by the establishment of second-generation antidepressant classes, including selective serotonin reuptake inhibitors (SSRIs) and serotonin-norepinephrine reuptake inhibitors (SNRIs)[2]. Although second-generation antidepressants have been shown to alleviate depression[3], treatment response is heterogeneous, and new side effect profiles have emerged (gastrointestinal, weight gain, sexual dysfunction). The degree of individual variation in the incidence and severity of these difficulties is high.

Treatment failure is commonly caused by the discontinuation of antidepressants from adverse side effects. Over half of individuals have been recorded to cease medication within the first six months of initial prescription[4]. Previously reported antidepressant adverse effects include sexual dysfunction[5–7], weight changes[8–11], insomnia[12–15], and suicidality[16–18]. However, these 'side effects' may also reflect ongoing symptoms of the depressive illness. For example, anhedonia is a cardinal symptom of major depressive disorder (MDD) which could explain lower levels of sexual interest and arousal leading to sexual function impairments[19,20]. Weight changes, sleep disturbances, and suicidality are also symptoms of depression[21] and its various phenotypic subtypes. Finally, other comorbid mental health disorders may amplify or trigger suicidal behaviours[22]. Whether these side effects stem from adverse reactions to antidepressants or whether they are extensions or exacerbations of characteristics of an individual's depression or a consequence of comorbidity with another disorder remains unclear.

Variability in medication response and tolerability may be inherited. For instance, genetic variation leading to changes in the function of antidepressant metabolising enzymes are believed to underlie side effects due to drug overexposure[23]. Five to seven percent of European ancestry individuals are estimated to be poor CYP2D6 metabolisers[24], one of the major metabolising enzymes of fluoxetine, paroxetine and fluvoxamine. Furthermore, variants in genes such as CYP2C19 and CYP3A4 have been linked to citalopram[25] and sertraline[26] differential metabolism and clearance. Metabolising enzymes are relevant hypotheses for understanding adverse side effects. Nonetheless, genetic variants within these enzymes have failed to reach significance in recent genome-wide association studies on treatment resistance[27] and response[28], suggesting that treatment outcomes might be more complex than previously thought. It is likely that genetic factors underlying antidepressant side effects are a product of drug-specific factors such as variation within drug-metabolising enzymes, as well as common (or non-drug-specific) factors, the nature of which remains elusive.

In general, the aetiology of antidepressant adverse side effects remains largely understudied. Thus, we aim to bridge this research gap by leveraging data from the Australian Genetics of Depression Study (AGDS) to gain insights into the prevalence, aetiology and genetic underpinnings of adverse side effects associated with antidepressant use. We investigate the prevalence and demographic risk factors for 23 side effects across ten commonly prescribed antidepressants. We test for SSRI or SNRI specificity and provide evidence for a co-occurring relationship between adverse side effects across different antidepressant medications. That is, participants who took two or more antidepressants were more likely to report the same side effects regardless of the antidepressant used. This co-occurrence would suggest a set of common risk factors underlie these side effects. Here, we use polygenic risk scores (PRS) to study the genetic aetiology of specific antidepressant adverse side effects to understand the nature of these common risk factors. PRS are an estimate of an individual's genetic risk for a given trait. They are calculated based on genome-wide association study (GWAS) results whereby genetic variants are linked to a trait of interest through an effect size (i.e., the increased risk per copy of the genetic variant). PRS are calculated in an independent sample by performing a sum of risk variants weighted by their effect size. PRS are gaining popularity due to their potential to enable many applications such as testing genetic overlap between traits, enabling risk stratification, and aiding diagnosis and personalised treatment[29]. We use PRS for MDD, BMI, insomnia and headaches to test for evidence of non-specific or shared genetic factors underpinning specific side effects. Overall, our results suggest drug exposure alone does not explain the occurrence of side effects, and a combination of specific and non-specific factors underlie their aetiology.

## Methods

**Sample recruitment and genotyping.** We use the Australian Genetics of Depression Study (AGDS), in which participants provide self-report responses on psychosocial factors of depression heterogeneity and antidepressant treatment outcomes ($N = 20,941$ with reported depression diagnosis) as well as DNA samples for genetic analysis. Sample recruitment has been described in detail elsewhere[30]. Briefly, 14.3% of volunteers were recruited by mail invitations distributed by the Australian Department of Human Services (DHS) and encouraged individuals who had previously used prescription antidepressants to participate in the last 4.5 years. Secondly, a nationwide media publicity campaign was broadcast. This campaign, targeted individuals who have sought medical attention from a psychiatrist or a psychologist for clinical depression. Recruited participants were directed to the study website to complete consent forms before answering the instruments. Once the instruments had been completed and informed consent for donation of a DNA sample was given, a GeneFix GFX-02 DNA extraction kit (Isohelix plc) was sent to participants to collect 2 mL of saliva for DNA extraction. Genotyping was performed using the Illumina Global Screening Array (GSA V.2.0.). Genotype data were cleaned by removing unknown or ambiguous map position, strand alignment, high missingness (>5%), deviation from Hardy–Weinberg equilibrium, low minor allele frequency (<1%) and GenTrain score <0.6 variants. Imputation was performed through the Michigan imputation server web service using the HRCr1.1 reference panel. Genotyped individuals were excluded from PRS analyses based on high genotype missingness, inconsistent and unresolvable sex or if deemed ancestry outliers from the European population, based on principal components derived from the 1000Genomes reference panel. The protocol for approaching participants through the DHS, enroling them in the study and consenting for all phases of the study (including invitation to future related studies) and accessing MBS and PBS records was approved by the Ethics Department of the Department of Human Services. The QIMR Human Research Ethics Committee also

approved all protocols for the ADGS data collection and scope for downstream studies under project number 2118. The study presented here falls within the scope of the analyses reviewed and approved under project 2118.

**Phenotype ascertainment**. This study focuses on participant-reported antidepressant adverse side effects. Participants first confirmed they had taken any of the ten most commonly prescribed antidepressants in Australia (sertraline, escitalopram, venlafaxine, fluoxetine, citalopram, desvenlafaxine, duloxetine, mirtazapine, amitriptyline and paroxetine). For each antidepressant taken, participants were asked whether they had experienced side effects and, if they did, to select which from a checklist with the twenty-three most commonly reported antidepressant side effects, including reduced sexual drive or desire, weight gain, dry mouth, nausea, drowsiness, insomnia, dizziness, fatigue, sweating, headache, suicidal thoughts, anxiety, agitation, shaking, constipation, diarrhoea, suicide attempt, blurred vision, muscle pain, vomiting, weight loss, runny nose and rash.

**Side effect correlations and structural equation modelling**. We used tetrachoric correlations as implemented in the psych library in R v3.6.1 to estimate the correlation (i.e. co-occurrence within the same set of people) of side effects across medications. Pairwise complete observations were used for these analyses. The correlation matrix was transformed into a distance matrix subjected to a minimum variance hierarchical clustering analysis using the *scipy* library in Python 3. The results are visualised with a clustergram generated using the *seaborn* and matplotlib libraries in Python 3.6. We further used structural equation modelling (OpenMx Rv3.6.2) to assess whether, for each side effect, there was evidence for drug-class-specific factors over and above a common factor. For each side effect, we fit a bifactor model consisting of a general factor loading onto the ten binary side effects, and two drug-class factors, "SSRI" and "SNRI" loading onto side effects from their respective drug class. The general factor is orthogonal to the drug-class factors. We refer to this model as the full model. Reduced models are also fit by removing the drug-class factors one at a time; these are the SSRI and SNRI models (Supplementary Fig. 3). Finally, a model consisting of a single general factor is also used for completeness. The four models are fit to the data using a full information maximum likelihood estimation assuming a liability threshold model for the binary manifest variables. After fitting, the simpler models are compared to the full model by the Akaike information criterion (AIC) and likelihood ratio test (LRT) with the mxCompare function implemented in OpenMx. Under this approach, the *p*-value represents whether a nested reduced model loses a significant amount of information compared to its full counterpart. Thus, a statistically significant *p*-value indicates that removing that drug-class factor results in a poorer fit.

**Genetic instruments and polygenic risk scoring**. To avoid biases due to population stratification and cryptic relatedness, only unrelated individuals of European ancestry were included in the genetic part of this study. PRS were calculated as a proxy for an individual's genetic liability to a trait. This study used publicly available GWAS results for depression[31], insomnia[32], chronic headaches[33], and BMI[34]. Genetic variant effect sizes were acquired from the GWAS data and used to calculate the predictive genetic risk for the traits investigated. Before estimating PRS, we excluded low ($r^2 < 0.6$) imputation quality and strand-ambiguous variants. We used two approaches to deal with correlation among genetic variants emerging through linkage disequilibrium (LD). First, we employed a recently developed

powerful method named SBayesR[35]. SBayesR estimates a conditional GWAS (i.e., one including all of the genetic variants as predictors simultaneously) using marginal GWAS summary statistics and LD measures between genetic variants (LD matrix) under a Bayesian multiple regression framework. This method has been shown to improve the polygenic prediction of complex traits. We also employed a more traditional clumping and thresholding procedure as sensitivity analyses. Briefly, PLINK (1.9)[36] was used to detect independent SNPs through a conservative clumping ($p1 = 1$, $p2 = 1$, $r^2 = 0.1$, kb=10,000) adjustment of linkage disequilibrium. Various *p*-value thresholds ($p < 5 \times 10^{-8}$, $p < 1 \times 10^{-5}$, $p < 0.001$, $p < 0.01$, $p < 0.05$, $p < 0.1$, $p < 0.5$, $p < 1$) were used to determine which variants to include for PRS calculation. Imputed genotype dosage data were used to calculate PRS by multiplying the variant effect size times the dosage of the effect allele. Finally, the total sum was calculated across all variants.

**PRS side effect association**. Logistic regressions were used to examine the association between participant-reported side effects and PRS. The regressions were adjusted for sex, age at study enrolment and the first 20 genetic principal components to further adjust for potential population stratification. Variance explained was calculated as Nagelkerke's pseudo $R^2$:

$$R^2 = \frac{1 - e^{\frac{2}{N}(LL_{null} - LL_{full})}}{1 - e^{\frac{2}{N}LL_{null}}}$$

where $LL_{full}$ and $LL_{null}$ are the log-likelihoods for the model with and without the PRS, respectively. Nominally significant results are defined as those with $p < 0.05$, and statistical significance was defined after Bonferroni correction for multiple testing. For the MDD analysis, we adjusted for the association of MDD PRS with the 25 side effects across medications ($p < 0.002$). For the other PRS, we adjusted for the testing of ten drugs ($p < 0.005$), and for the sensitivity analyses (using clumping and thresholding), we adjusted for eight thresholds times ten medications ($p < 0.000625$). This method is relatively conservative, as it does not account for the moderate to high co-linearity within the eight PRS and within the side effects across medications. Results are visualised as heat maps of variance explained using *seaborn* and *matplotlib* in Python 3.6. These analyses were performed using complete case data. Scripts and data for this study, including PRS (i.e. SBayesR effect sizes) are available online at doi: 10.5281/zenodo.5533372.

**Reporting summary**. Further information on research design is available in the Nature Research Reporting Summary linked to this article.

## Results

**Demographics and side effect prevalence**. As previously reported[30], the majority of AGDS participants were female (75%). The average age was 43 (s.d. = 15.3) years old. Most people (60–75%) reported at least one side effect regardless of antidepressant taken (Supplementary Fig. 1). Figure 1 shows the prevalence of reported side effects for males and females across the ten studied antidepressants. Reduced sex drive and weight gain had the highest prevalence. The least prevalent side effects were suicide attempt, blurred vision, rash, weight loss, and muscular pain. Overall, there were significant differences between the prevalence of side effects for males and females across all medications (Table 1 and Supplementary Data 1). Males reported experiencing reduced sex drive or function more often than females.

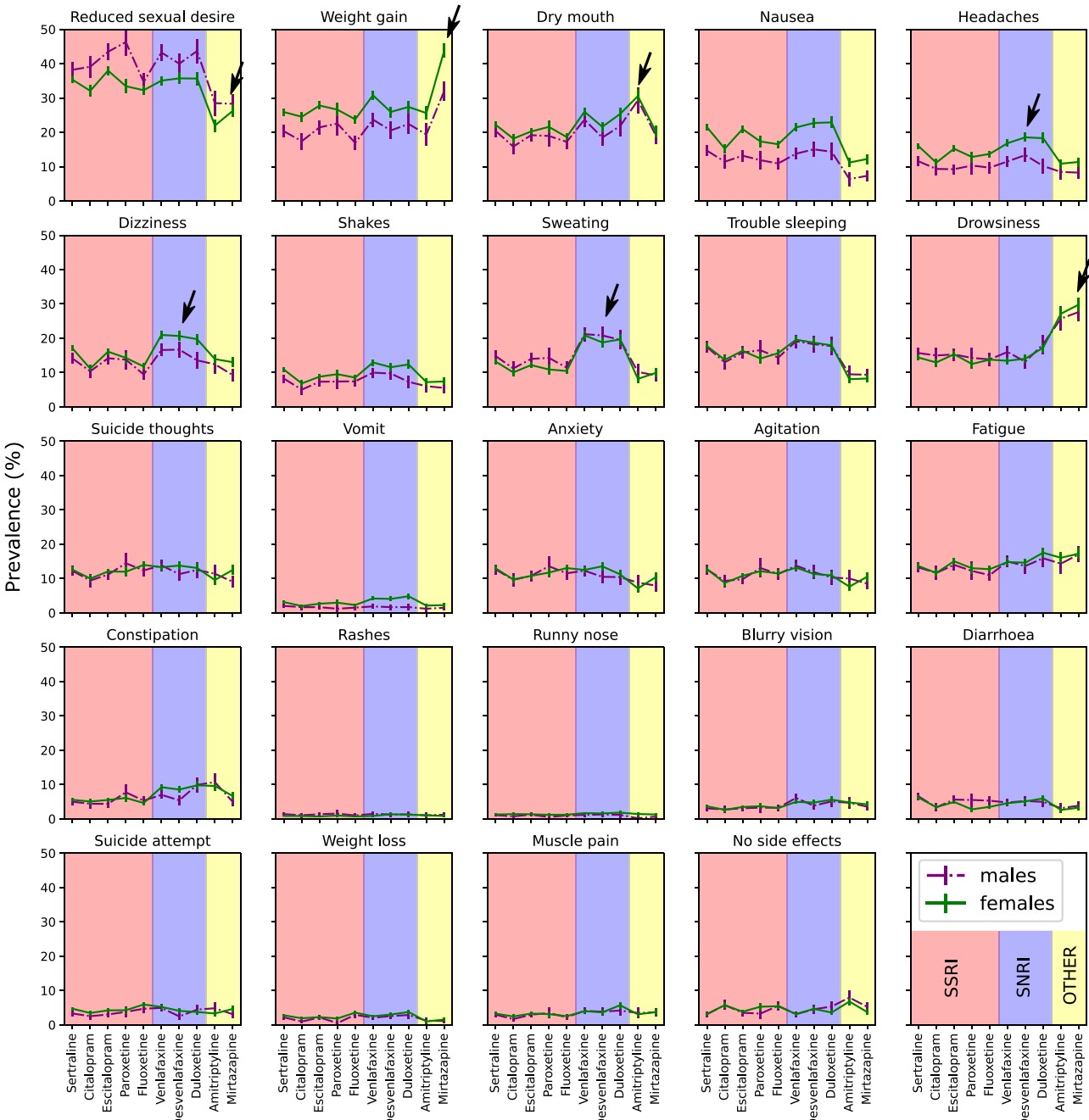

**Fig. 1 Side effect prevalence across medications.** Plots depicting the prevalence and 95% confidence interval for each of the 25 studied side effects across ten medications. Results are split by males (dotted lines) and females (solid lines). The plots are shaded according to antidepressant drug class. Some of the findings discussed in the results are highlighted by black arrows. *p*-values are available in Supplementary Data 1. SSRI-Selective serotonin reuptake inhibitors;SNRI—serotonin and norepinephrine reuptake inhibitors.

Conversely, women were more likely to report weight gain compared to males (Supplementary Data 1). Other side effects showing a robust difference (i.e., a difference observed for five or more medications) in prevalence between males and females included nausea, headaches, dizziness, shakes and vomits. Antidepressants are typically prescribed in a specific order. To adjust for the possibility that the course of illness underlies these side effects (rather than the drug class), we performed a sensitivity analysis re-estimating the side effect prevalence using a subsample of participants who reported only taking one antidepressant. These results were highly consistent with the full sample analysis (Supplementary Fig. 2).

Overall, participants reporting specific side effects were younger than those not doing so. Conversely, two side effects, rashes and runny nose, were reported by older participants (Table 1). Some side effects such as dizziness, headaches and sweating showed a higher prevalence across SNRIs than SSRIs. These patterns were sometimes sex-specific (Fig. 1 and Supplementary Data 1). We observed a higher prevalence of weight gain from mirtazapine, dry mouth from amitriptyline and drowsiness from both amitriptyline and mirtazapine than SNRIs and SSRIs. Conversely, reduced sexual desire or function was more commonly reported for SSRIs or SNRIs than amitriptyline and mirtazapine (Fig. 1).

**Table 1 AGDS demographics and side effect prevalence across medications.**

| | Males N | Females N | Sex p-value[a] | Endorsed side effect age (s.d.) | Not endorsed side effect age (s.d.) | Age p-value[b] |
|---|---|---|---|---|---|---|
| N | 5111 | 15830 | — | — | — | — |
| Age mean (s.d.) | 47.99 (15) | 41.41 (14) | 6.6e−160 | — | — | — |
| Reduced sexual desire | 2251 (44%) | 6264 (39%) | 1.5e−08 | 41.0 (13.98) | 44.4 (16.01) | 1.1e−58 |
| Weight gain | 1402 (27%) | 5695 (35%) | 3.2e−29 | 41.9 (13.89) | 43.6 (15.96) | 1.1e−13 |
| Dry mouth | 1236 (24%) | 4544 (28%) | 3.2e−10 | 42.7 (14.20) | 43.1 (15.71) | 0.071 |
| Nausea | 867 (16%) | 4352 (27%) | 1e−51 | 37.7 (13.34) | 44.8 (15.52) | 1.6e−187 |
| Headaches | 704 (13%) | 3282 (20%) | 3.1e−28 | 38.3 (13.74) | 44.1 (15.45) | 2e−106 |
| Dizziness | 959 (18%) | 3930 (24%) | 5.2e−19 | 38.4 (13.43) | 44.4 (15.57) | 5.3e−130 |
| Shakes | 571 (11%) | 2466 (15%) | 7.4e−15 | 38.8 (14.22) | 43.7 (15.37) | 3.8e−62 |
| Muscle pain | 234 (4%) | 837 (5%) | 0.045 | 42.0 (14.98) | 43.1 (15.33) | 0.029 |
| Sweating | 997 (19%) | 3291 (20%) | 0.048 | 40.3 (13.75) | 43.7 (15.61) | 6.8e−38 |
| Vomit | 147 (2%) | 826 (5%) | 4.7e−12 | 35.7 (12.65) | 43.4 (15.34) | 6e−53 |
| Constipation | 395 (7%) | 1489 (9%) | 2.70E−04 | 43.3 (15.01) | 43.0 (15.34) | 0.440 |
| Diarrhoea | 368 (7%) | 1176 (7%) | 0.590 | 39.9 (13.87) | 43.3 (15.39) | 8.8e−17 |
| Drowsiness | 1173 (22%) | 3709 (23%) | 0.480 | 39.7 (14.30) | 44.0 (15.46) | 6.8e−67 |
| Trouble sleeping | 1052 (20%) | 3672 (23%) | 1.00e−04 | 39.6 (14.35) | 44.0 (15.43) | 2.3e−70 |
| Anxiety | 794 (15%) | 2973 (18%) | 1.5e−07 | 39.1 (14.24) | 43.9 (15.40) | 1.6e−68 |
| Agitation | 786 (15%) | 2816 (17%) | 7.2e−05 | 39.6 (14.12) | 43.7 (15.45) | 1.3e−50 |
| Fatigue | 912 (17%) | 3181 (20%) | 4.20e−04 | 39.4 (14.49) | 43.9 (15.38) | 8.4e−63 |
| Weight loss | 157 (3%) | 795 (5%) | 5.9e−09 | 36.1 (13.42) | 43.3 (15.32) | 5.1e−47 |
| Rashes | 97 (1%) | 257 (1%) | 0.190 | 45.5 (14.96) | 43.0 (15.31) | 0.002 |
| Runny nose | 82 (1%) | 344 (2%) | 0.012 | 44.9 (15.64) | 43.0 (15.30) | 0.010 |
| Blurry vision | 266 (5%) | 1012 (6%) | 0.002 | 42.6 (14.60) | 43.0 (15.35) | 0.280 |
| Suicide thoughts | 699 (13%) | 2560 (16%) | 1.9e−05 | 38.2 (14.24) | 43.9 (15.33) | 4.7e−87 |
| Suicide attempt | 248 (4%) | 1090 (6%) | 2.4e−07 | 35.9 (13.41) | 43.5 (15.31) | 4.5e−69 |
| Other side effects | 632 (12%) | 1968 (12%) | 0.900 | 41.3 (14.05) | 43.3 (15.47) | 1.3e−09 |
| No side effects | 334 (6%) | 1140 (7%) | 0.110 | 43.8 (15.04) | 43.0 (15.33) | 0.044 |

[a]Two sample Z proportion test.
[b]Two sample t-test. Data of each side effect per medication studied are available in Supplementary Data 1.

**Side effects co-occur across medications.** For each antidepressant taken, participants reported whether or not they had experienced specific side effects. Some individuals had taken more than one antidepressant. Thus, several subsets of participants with overlapping data for different antidepressants were available. This overlap enabled us to assess whether the same side effects occur independently across different antidepressants or whether they co-occur (covary) across medications. We identified high correlations for the same side effects across medications (within side effect median correlation = 0.57, within medication median correlation = across side effect median correlation = 0.27 see Supplementary Fig. 3). A clustering analysis based on these correlations visually grouped these variables by side effect rather than antidepressant or medication class (Fig. 2). Overall, these results would suggest that antidepressant side effects co-occur in the same people regardless of the medication of choice or their class, which would be consistent with the existence of a set of general non-specific (i.e. independent of drug mode of action) factors partly underlying their aetiology. Structural equation models were used to compare a bifactor model with common and drug-class factors against reduced models, including only specific drug classes or no drug class (Supplementary Fig. 4a). Our results suggested that the common factor captured a high proportion of the covariance across medications. A single common factor model was preferred over the SSRI, SNRI, or the full model for some side effects. However, we identified evidence for medication-class-specific effects for other side effects. For example, sweating, insomnia, suicide thoughts, and suicide attempts showed evidence of an SNRI-specific factor and vomits nausea, and drowsiness showed evidence of an SSRI-specific factor. In contrast, dizziness and constipation showed evidence for both factors over and above a general factor (Supplementary Fig. 4b and Supplementary Data 2).

**Genetic factors underlie side effects.** We employed a polygenic risk scoring strategy to assess whether the general factors identified have a genetic basis (Fig. 3a; see "Methods"). Depression PRS were positively associated with most side effects when pooling data across medications. The side effects that showed the strongest association with depression PRS were suicide thoughts and suicide attempt, which is consistent with their intricate relationship with depression (Fig. 3b; Table 2). The strength of these associations was reduced, and increased heterogeneity was observed when splitting per antidepressant regardless of what PRS method was used (Supplementary Fig. 5). Only dizziness, constipation, agitation, suicide thoughts, suicide attempt and reduced sexual desire, showed significant association with depression PRS when splitting per antidepressant (Supplementary Fig. 5; Supplementary Data 3).

We then tested whether polygenic risk for related common traits underlies the risk for three specific side effects. We chose to study weight gain, insomnia, and headaches to identify related complex traits (BMI, insomnia, and headaches, respectively) for which well-powered GWAS data is readily available. BMI PRS were strongly and robustly associated with weight gain across all medications (Fig. 4; Supplementary Data 4). PRS for headaches and insomnia showed evidence of association with headaches and trouble sleeping as side effects. PRS for headaches were associated with headaches from sertraline and, to a lesser extent, with headaches from venlafaxine. Insomnia PRS showed suggestive evidence predicting insomnia from escitalopram and amitriptyline (Supplementary Fig. 6; Supplementary Data 5-6). Amitriptyline is sometimes used to treat insomnia. We identified a higher insomnia PRS on average for participants that reported taking amitriptyline (Supplementary Fig. 7). To assess whether this could be confounding the association, we repeated the analysis

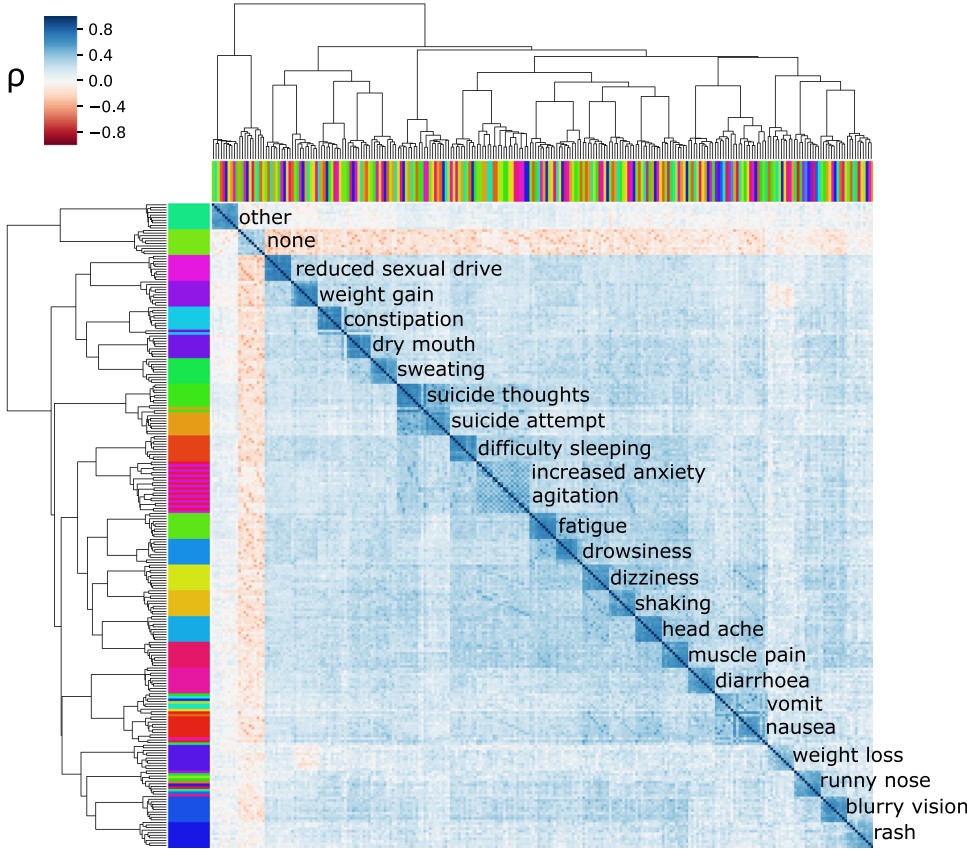

**Fig. 2 side effects co-occur across medications.** Results of hierarchical clustering based on tetrachoric correlations of the side effects across drugs. The colour shows the correlation (co-occurrence) of patient-reported side effects. The top and side colour bars represent the ten medications and the 25 studied side effects, respectively. Labels show the most abundant side effect in each cluster. $\rho$- tetrachoric correlation coefficient.

adjusting for whether participants reported taking amitriptyline for insomnia. Insomnia PRS was associated with participants taking amitriptyline for insomnia (OR = 1.2 95%C.I. = [1.1–1.3]). Nonetheless, whether participants took amitriptyline for insomnia was not associated with insomnia as a side effect from this medication (OR = 1.2 95%C.I. = [0.8–1.8]). As expected, insomnia PRS was still predictive of insomnia as a side effect from amitriptyline after adjusting for whether it was taken for insomnia or not (OR = 1.3 95%C.I. = [1.1–1.5]). Overall, the direction of effects between PRS and side effects were positive for all medications (e.g., higher BMI PRS higher risk of weight gain as a side effect). Still, the errors varied potentially due to sampling size differences.

## Discussion

In this study, we aimed at gaining insights into the genetic aetiology of self-reported antidepressant adverse side effects. Our study has several insights, including (1) providing prevalence estimates for side effects outside of a controlled clinical trial; (2) assessing drug-class specificity of antidepressant side effects; (3) testing and providing evidence for non-specific factors (i.e. not related to the type of medication), which is of particular interest for some side effects such as suicidality; and 4) testing for genetic factors underlying the aetiology of side effects using PRS. We identified the most common side effect to be reduced sexual drive or function, followed by weight gain. Reduced sexual function was most prevalent among males taking paroxetine, an SSRI, whereas weight gain was the most prevalent among females taking amitriptyline, a tricyclic antidepressant (TCA). These findings are consistent with research findings showing SSRIs

exhibit the most adverse sexual effects[37], whereas TCAs have been established to cause weight increases[38].

Studies have suggested that SSRI-based sexual dysfunction may result in 40–65% of individuals ceasing treatment[39]. It is hypothesised that testosterone and dopamine neurotransmitters are dysregulated by SSRIs, a plausible hypothesis considering the role that testosterone plays in sexual function and the high amounts circulating in males compared to females[40,41]. Moreover, serotonin plays a crucial role in initiating smooth muscle contraction of the genito-urinary system and regulating the response of the sexual cycle; thus, exogenous substances such as SSRIs that alter these mechanisms may cause sexual dysfunction in both males and females[42]. SNRIs have been reported to lead to more *dizziness*, *trouble sleeping* and *dry mouth* than SSRIs[38,43]. We found a higher prevalence for dizziness and sweating from venlafaxine, desvenlafaxine and duloxetine (SNRIs) compared to other antidepressants.

SSRIs are generally prescribed as first-line agents due to their safety profile, evident since clinical studies from the 1980s onwards, for individuals with multiple comorbidities[44–46]. Moreover, as they are better tolerated than older agents, this results in more effective long-term management[47]. While we observed specific instances where side effects were more prevalent across other antidepressant types, we did not find a lower prevalence for *any side effect* nor a higher prevalence of *no side effects* for SSRIs. This might be explained by the fact that non-specific factors (i.e., regardless of medication class) seemed to underlie the self-reporting of side effects.

Side effects co-occurred across all antidepressants assessed despite their variable modes of action and metabolism[48]. Participants who reported a side effect for one medication were more

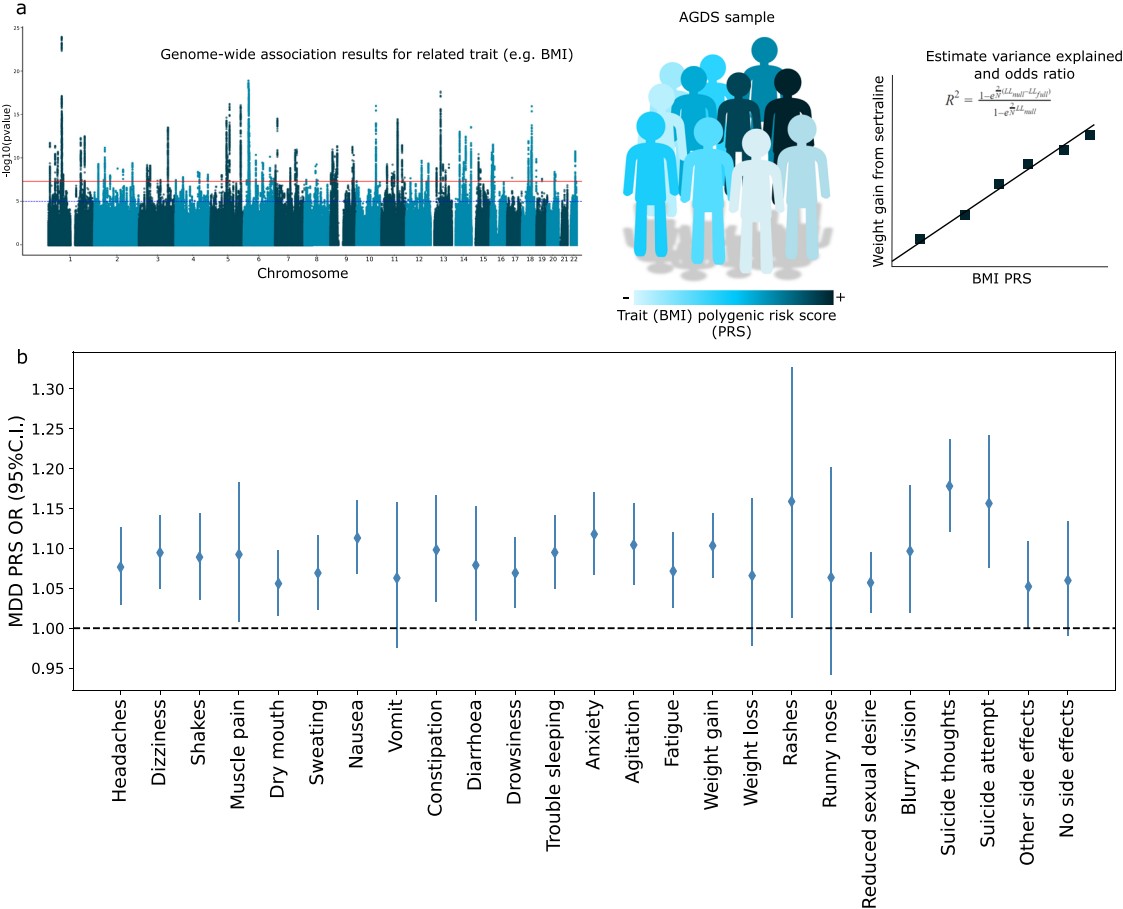

**Fig. 3 Genetic factors underlying side effects. a** Overview of the polygenic risk scoring approach employed. Results from a published GWAS are used to estimate the polygenic risk for a given trait in the AGDS. These scores can be used to predict the side effect of interest. **b** Forest plot showing the results, odds ratio (OR) and 95% (C.I.) for the association between MDD PRS and side effects across antidepressants. MDD—major depressive disorder; PRS—polygenic risk score.

likely to report that same side effect for other medications. This strongly suggests the existence of common non-specific factors, which are potentially a mixture of shared pharmacological and genetic factors underlying their aetiology. Our structural equation modelling analyses suggested that for some side effects, a common factor model would be preferred over models considering the drug classes.

The evidence for lack of drug-class factors was not equally strong for all side effects. For example, *sweating* showed the best fit to a model containing an SNRI factor along with the common factor. This is consistent with the higher prevalence of sweating reported for SNRIs compared to other drugs, which would already imply an SNRI-specific factor increasing its prevalence. For some side effects, common factors are further supported by the fact that they are widely reported from other types of medications; for example, *weight gain* is a common side effect not only for antidepressants but also for antipsychotics, antihyperglycemics, antihypertensives and corticosteroids[49]. The nature of these common factors is complex and might include a mixture of a *nocebo* effect[50], shared metabolism, common environmental and genetic factors.

This study uniquely addressed the role of genetic factors on antidepressant adverse side effects. First, we evaluated whether the genetic liability to depression, which has been linked to increased depression severity, recurrence and persistence[51], is associated with antidepressant side effects. Depression PRS were associated with many side effects, particularly those that could be

considered depression symptoms or common comorbidities such as anxiety, trouble sleeping and suicidal behaviours. When testing across individual antidepressants, a more heterogeneous pattern was observed. Although we cannot rule out reduced power due to subsampling when performing these analyses, the heterogeneity could imply that interactions between specific drugs and depression PRS underlie the studied adverse side effects. For example, MDD PRS was robustly associated with increased suicidality for venlafaxine, but evidence for association with other drugs did not reach statistical significance. This result is not easily attributed to power given that sertraline, and not venlafaxine, is the drug for which we have the largest sample size. Furthermore, venlafaxine has been associated with increased suicidality compared to placebo in modern meta-analyses[52,53]. This is also consistent with our findings of an SNRI factor underlying suicide thoughts over and above a general factor. Our results suggest that this increased risk might be mediated by genetic factors, which opens up the opportunity for genetic risk stratification, management and genetically informed therapies.

The second type of factor consisted of liability to traits related to the nature of specific side effects. We tested for the association between PRS for BMI, chronic headaches and insomnia with weight gain, headaches and trouble sleeping as side effects from antidepressants. We found strong evidence for BMI PRS predicting weight gain for all medications. Evidence for headaches and insomnia was moderate, and associations remained heterogeneous across medications and PRS. These observations are

**Table 2 Polygenic prediction of specific side effects across medications.**

| Antidepressant | BMI PRS predicting weight gain | | | Insomnia PRS predicting trouble sleeping | | | Headaches PRS predicting headaches | | |
|---|---|---|---|---|---|---|---|---|---|
| | OR (95%C.I.) | P-value | Variance explained (%) | OR (95%C.I.) | P-value | Variance explained (%) | OR (95%C.I.) | P-value | Variance explained (%) |
| Sertraline | 1.24 (1.17–1.32) | **4.66e-12** | 1.25 | 1.06 (0.99–1.14) | 0.077 | 0.09 | 1.16 (1.08–1.25) | **8.17e-05** | 0.49 |
| Escitalopram | 1.21 (1.13–1.29) | **3.58e-08** | 1.01 | 1.12 (1.03–1.21) | 0.008 | 0.27 | 1.06 (0.97–1.16) | 0.178 | 0.08 |
| Venlafaxine | 1.20 (1.12–1.28) | **3.72e-07** | 0.93 | 1.03 (0.95–1.11) | 0.463 | 0.02 | 1.11 (1.01–1.21) | 0.025 | 0.22 |
| Amitriptyline | 1.25 (1.12–1.40) | **9.84e-05** | 1.38 | 1.28 (1.07–1.52) | 0.007 | 1.03 | 1.09 (0.93–1.28) | 0.268 | 0.15 |
| Mirtazapine | 1.20 (1.09–1.32) | **2.03e-04** | 0.95 | 1.03 (0.88–1.20) | 0.745 | 0.01 | 1.04 (0.90–1.20) | 0.560 | 0.04 |
| Desvenlafaxine | 1.22 (1.11–1.33) | **3.01e-05** | 1.03 | 1.13 (1.03–1.25) | 0.015 | 0.38 | 1.08 (0.98–1.20) | 0.126 | 0.16 |
| Citalopram | 1.37 (1.24–1.50) | **9.06e-11** | 2.5 | 1.08 (0.96–1.21) | 0.194 | 0.12 | 1.11 (0.98–1.26) | 0.088 | 0.23 |
| Fluoxetine | 1.26 (1.17–1.36) | **5.75e-09** | 1.42 | 1.03 (0.94–1.13) | 0.523 | 0.02 | 1.02 (0.93–1.12) | 0.689 | 0.01 |
| Duloxetine | 1.20 (1.08–1.32) | **3.60e-04** | 0.94 | 1.14 (1.02–1.28) | 0.026 | 0.4 | 1.03 (0.92–1.15) | 0.626 | 0.02 |
| Paroxetine | 1.22 (1.09–1.36) | **6.28e-04** | 1.1 | 1.16 (1.01–1.33) | 0.042 | 0.47 | 1.16 (1.00–1.35) | 0.053 | 0.46 |

Results from logistic regressions predicting weight gain, trouble sleeping and headaches using BMI, Insomnia and headaches PRS respectively. Bolded p-values represent values significant after multiple testing correction ($p < 0.005$). Results shown for SBayesR, for clumping and thresholding sensitivity results see Supplementary Data.

consistent with weight gain being a potentially less specific side effect. As discussed above, weight gain is frequently reported for very distinct classes of drugs.

On the other hand, it is unclear whether the heterogeneity observed for headaches and insomnia is due to statistical power. The significant association for headaches was under sertraline, the most reported drug in our sample, which would be consistent with a lack of power underlying the lack of significance for other similar (SSRI and SNRI) pharmacological medications. Genetic risk for insomnia was nominally associated with insomnia from amitriptyline. TCAs are generally associated with drowsiness and fatigue, but people with a higher genetic risk for insomnia were more likely to report insomnia as a side effect from amitriptyline. This may represent a prescribing bias caused by the fact that TCAs are often used (in lower doses) to treat insomnia. We performed a sensitivity analysis by adjusting for whether participants reported taking amitriptyline for insomnia and showed that this did not confound the association we observed. Thus, our observations may be evidence of the genetic risk for insomnia *nullifying* the energy-lowering effects of amitriptyline. Altogether, our results would suggest that in addition to genetic factors associated with depression, the genetic liability to side effect-related traits, such as BMI for weight gain, also underlie their aetiology. These results further prove the principle of using genetic data to study and predict antidepressant side effects. Genetics-driven prediction of treatment outcomes is one of the major challenges towards achieving precision medicine.

This study represents one of the most powered and comprehensive explorations of antidepressant side effects. The unprecedented detail and sample size of the AGDS enable us to gain valuable insights into the genetic and pharmacological underpinnings of antidepressant side effects. Nonetheless, certain limitations need to be acknowledged. First, the retrospective nature of this study and the reported phenotypes are prone to recall bias and subjective definition of a side effect. Second, individual side effect prevalence was estimated across the whole sample, including participants that have taken more than one medication. As such, the non-independence between these estimates should be considered when comparing side effect prevalence across medications. We tackled this limitation by performing a sensitivity analysis, focusing on participants who had taken only one antidepressant; this approach showed highly concordant results. Participants might also have taken non-antidepressant medications in combination with antidepressants. We did not have sufficiently detailed data to assess or adjust for this. We also did not collect information on the antidepressant dosages nor on more serious, albeit rarer, side effects such as the onset of mania, attempted suicide resulting in hospitalisation or myocardial infarction. Our genetic analyses were performed on a subset of individuals of European ancestry to prevent spurious associations arising from population stratification. Thus, caution must be taken when generalising our PRS findings to populations of a distinct genetic background.

In conclusion, we characterised the aetiology of side effects in a sample of Australian adults who reported depression over their lifetime. *Sexual dysfunction* and *weight gains* are the most commonly reported side effects. Some side effects, including weight gain and sexual dysfunction, showed clear differential sex-specific prevalence. We used clustering and structural equation modelling to test for co-occurrence and drug-class specificity of the side effects. We observed that side effects significantly co-occurred across medications, suggesting shared pharmacological or genetic factors underlie their aetiology. As such, these reported side effects may be manifestations of depression severity, persistence, recurrence or non-response to treatment or its comorbidities. We employed PRS to test whether these shared factors had a genetic

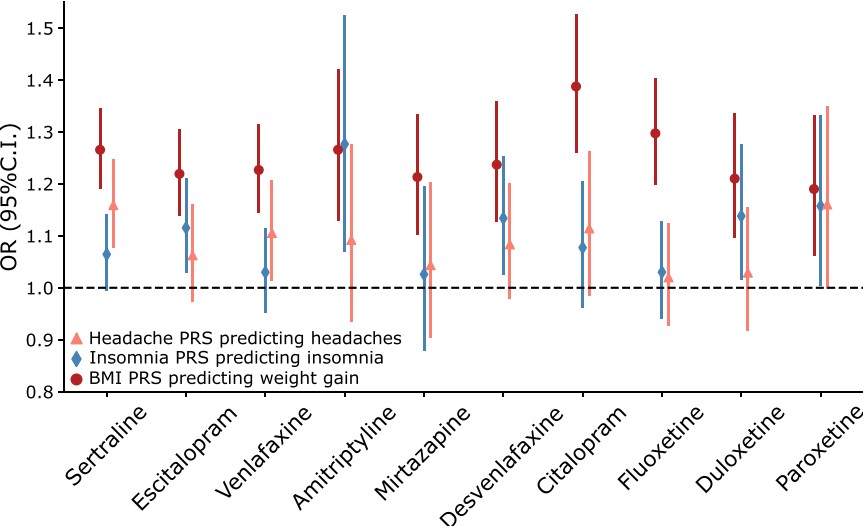

**Fig. 4 Genetic prediction of antidepressant side effects.** Forest plots show the results of associations between polygenic risk scores for BMI, insomnia and chronic headaches predicting weight gain, insomnia and headaches reported as side effects for antidepressant medications. Markers represent odds ratios (OR), and error bars represent 95% confidence intervals (C.I.).

component. Depression PRS were associated with most of the side effects studied, strongly suggesting that a depression severity factor underpins reporting of some side effects. Furthermore, trait-specific PRS, such as BMI, were predictive of related side effects such as weight gain. Altogether, our results suggest that drug exposure alone does not explain the occurrence of side effects, and a combination of specific and non-specific factors underlie their aetiology. Future studies on the genetic aetiology (e.g. performing well-powered GWAS) of adverse side effects will enable further insights into their underpinnings as well as the possibility of genetically driven risk stratification and adverse side effect prophylaxis strategies.

## Data availability
Summary data on prevalence and effects described in this manuscript is available in the supplementary data. Source data for the main figures of the manuscript is available in supplementary data 1–6 or online https://doi.org/10.5281/zenodo.5533372[54]. GWAS summary statistics used in this study are publicly available. For MDD the data were obtained from the PGC leaving out the QIMR cohorts. The full results are available at https://www.med.unc.edu/pgc/download-results/. The insomnia summary statistics are available at https://ctg.cncr.nl/software/summary_statistics. The BMI summary statistics are available at https://portals.broadinstitute.org/collaboration/giant/index.php/GIANT_consortium_data_files. The chronic headaches summary statistics are available online at online https://doi.org/10.5281/zenodo.5533372. Furthermore, all of the effect sizes for PGS are available online at the PGS Catalogue under publication ID PGP000238. Access to the AGDS data is restricted due to the ethical guidelines governing the study, but may be accessible following ethical review and data transfer agreements, please contact Nicholas Martin (nick.martin@qimrberghofer.edu.au) with any queries related to accessing AGDS data.

## Code availability
This study used python (3.6) and R (3.6.2) code for data analysis. Code is available as HTML notebooks or R scripts online at https://doi.org/10.5281/zenodo.5533372[54].

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

## Acknowledgements
Data collection for AGDS was possible thanks to funding from the Australian National Health & Medical Research Council (NHMRC) to NGM, NRW, SEM, IHB, EMB (GNT1086683) and Medical Research Future Fund (APP1200644). We thank our colleagues Richard Parker, Simone Cross, Scott Gordon and Lenore Sullivan for their valuable work coordinating all the administrative and operational aspects of the AGDS project. AIC is supported by a UQ Research Training Scholarship from The University of Queensland (UQ). MER thanks the support of NHMRC and the Australian Research Council (ARC) through an NHMRC-ARC Dementia Research Development Fellowship (GNT1102821). SEM is supported in part by NHMRC investigator grant APP1172917. The views expressed are those of the authors and not necessarily those of the affiliated or funding institutions.

## Author contributions
A.I.C., N.G.M., S.E.M., N.R.W., I.B.H., E.M.B. and M.E.R. designed this study. A.I.C. performed the analyses. A.I.C. and A.M. wrote the first version of the manuscript. A.I.C. and J.G.T. performed the SEM analyses. N.G.M., S.E.M., N.R.W., E.M.B., P.L. and I.B.H. designed and directed the AGDS data collection efforts. All authors contributed to the interpretation of the results and provided feedback on the preliminary versions of the manuscript.

## Competing interests
I.B.H. has been: Commissioner of Australia's National Mental Health Commission (2012–2018); Co-director of Health & Policy at the Brain & Mind Centre, University of Sydney; leading community-based and pharmaceutical industry-supported projects (Wyeth, Eli Lilly, Servier, Pfizer, AstraZeneca) focused on the identification and better management of anxiety and depression; a member of the Medical Advisory Panel for Medibank Private until October 2017; a board member of Psychosis Australia Trust; a member of the Veterans Mental Health Clinical Reference Group; and Chief Scientific Advisor to and an equity shareholder in Innowell. A.I.C., A.M., N.G.M., J.G.T., S.E.M., P.L., J.G.T., N.R.W., E.M.B. and M.E.R. have nothing to disclose.
