## [Peer Review File · Communications Medicine]

Reviewers' comments:

Reviewer #1 (Remarks to the Author):

Campos et al gave an insightful and clear overview of a large dataset on side effects of antidepressant use. They found that the influence of drug type was minimal on the common factors of side effects. PRS for various phenotypes, such as insomnia, BMI and headache, were associated with reported side effects, and differences were found across a wide range of antidepressants. The study can benefit from a few sensitivity analyses and more extensive report of model statistics.

1. As the authors mentioned that SSRIs are often prescribed first before switching to other types of antidepressants, would the results for some of the side effects related to other types of medications (e.g. SNRIs) driven by the course of disease? One way to investigate this is to covary for the number of medications taken so far, if this type of data is available.

2. How did the authors handle multiple medications taken at the same time? Is it likely that some of the side effects were caused by a combination of antidepressants taken?

3. In the SEM models (reported in the main text and Supplementary Figure 2), were the correlations between medications included? It is not clear either in the text or in the figure. If not, a sensitivity analysis including the correlations may be necessary.

4. On page 8, there are few statistics reported regarding the SEM models in the 'Side effects co-occur across medications' section. Many statements were not supported by statistical evidence. For example, 1) in 'we identified high correlations for side effects across medications', a range of correlation coefficients should be given; 2) in 'A clustering based on these correlations grouped these variables by side effect rather than by antidepressant or medication class', model statistics should be given to support the conclusion for model comparison.

5. In addition to point 4, only AICs were reported in Supplementary Table 2 for SEM models. I encourage the authors to include other measures such as CFI, TLI, RMSEA. Similarly, it would be necessary to report association strengths in Supplementary Figure 2 or at least a range of effect sizes for parallel tests.

Reviewer #2 (Remarks to the Author):

The paper is an investigation into the effects of genetic loading for depression, BMI, insomnia and headaches on self-reported anti-depressant side effects. They used an Australian depression cohort that had details on participant's genetics, class of antidepressants taken, as well as a list of 23 side effects. The paper makes use of both structural equation modelling and sophisticated PRS generating software (SBayesR) which is a major plus point of the manuscript. The paper concludes that antidepressant side effects could well be driven not just by class of antidepressant but in combination with an individual's genetic loading for the tested traits.

I think that this paper is a useful addition to the current literature in a very important area of research. The paper is concise, written with a sufficient level of detail for other to reproduce the analyses and the conclusions are well thought out. However, I do not think the paper is suitable for publication in its current form without 1 major change to the methodology. For BMI and chronic headaches, summary statistics from Neale's public repository of UK Biobank GWAS statistics were used. The use of these outputs can be useful but as they are generated using an automated pipeline they have minimum data cleaning, and as such, should not be used as a basis for a publication if other summary statistics are available. The analysis should be rerun using more appropriate summary statistics for BMI and chronic headaches. Here are two potential papers that would be

more appropriate.

Yengo et al, BMI GWAS/meta-analysis

<https://www.ncbi.nlm.nih.gov/pmc/articles/PMC6488973/>

Gormley et al, migraine GWAS

<https://www.ncbi.nlm.nih.gov/pmc/articles/PMC5331903/>

A two minor comments:

- Although 23 side effects were investigated there were only 5 risk scores created. The paper should go into more detail as to why those particular traits were selected to create risk scores and not at genetic loading of any of the other side effects.
- Figure one has lines connecting the data points, but the x axis scale is categorical. The plots could be made clearer by removing the line connecting the data points and using box plots instead with males and females side by side for each category.

Joey Ward

Reviewer #3 (Remarks to the Author):

This is an interesting paper, however it is not clear what is really new, what is the take home message (about the relationship between individual antidepressants and specific adverse events) and how the authors validated their findings (see limitations in the discussion section).

There sample size of the controls is relatively very small; moreover, the retrospective design of this study and the recollection bias (as reported by the authors themselves) are important concerns that limit the reliability of these findings.

Many of the references are old or not the most appropriate/relevant.

I think this article would benefit from the replication of the findings in a different cohort and from the inclusion of clinical researchers with expertise in antidepressant studies among the co-authors.

There is no protocol, so it is not clear why and how the authors selected the list of side effects.

Tables and figures should be more informative

We thank the reviewers for their insights, comments and feedback. Below we provide a point by point response.

Reviewer #1 (Remarks to the Author):

Campos et al gave an insightful and clear overview of a large dataset on side effects of antidepressant use. They found that the influence of drug type was minimal on the common factors of side effects. PRS for various phenotypes, such as insomnia, BMI and headache, were associated with reported side effects, and differences were found across a wide range of antidepressants. The study can benefit from a few sensitivity analyses and more extensive report of model statistics.

We thank the reviewer for their time assessing our manuscript and for the constructive feedback.

1.As the authors mentioned that SSRIs are often prescribed first before switching to other types of antidepressants, would the results for some of the side effects related to other types of medications (e.g. SNRIs) driven by the course of disease? One way to investigate this is to covary for the number of medications taken so far, if this type of data is available.

We thank the reviewer for this insightful question. Following guidelines, the pattern the reviewer suggests would be a common one. As such, it is true that the course of disease, rather than SNRI pharmacological effects, might underlie the results. Nonetheless, our data showed a subset of participants who reported taking only one antidepressant. Using this subsample, it is possible to perform a sensitivity analysis for the course of illness:

Full analysis

Sensitivity analysis

As expected, the error increased due to the reduced sample sizes. Nonetheless, we observe a similar pattern to that observed in the original analysis (see the arrows that highlight the results in the left panel). We also note this approach should be done with caution as participants who report taking only one antidepressant may be a combination of people who recently started taking antidepressants or who responded very well to their first treatment. Thus, we have included this new result in the supplementary material and mentioned it in the results section.

2. How did the authors handle multiple medications taken at the same time? Is it likely that some of the side effects were caused by a combination of antidepressants taken?

The reviewer is correct. We believe the sensitivity analysis showed above also tackles this issue. The sensitivity analysis focused only on a subset of participants reporting taking a single antidepressant. Nonetheless, we did not gather data on other concurrent (non-antidepressant) medications with sufficient detail. Thus, we have added a sentence in the limitations section acknowledging this possibility:

*"Second, side effect prevalence were estimated across the whole sample, including participants that have taken more than one medication. As such, the non-independence between these estimates should be taken into account when comparing side effect prevalence across medications. We tackled this limitation by performing a sensitivity analysis that showed highly concordant results. **Nonetheless, participants might also have taken non-antidepressant medications in combination with antidepressants. We did not have sufficiently detailed data to assess or adjust for this.**"*

3. In the SEM models (reported in the main text and Supplementary Figure 2), were the correlations between medications included? It is not clear either in the text or in the figure. If not, a sensitivity analysis including the correlations may be necessary.

The correlation between medications is modelled and partitioned into several factors (depending on the model tested) in our analysis. We cannot fit a model with the residual covariance across medications as this would be unidentified. We think the reviewer likely refers to models that allow the drug-class factors (SNRI and SSRI) to correlate. We have repeated this analysis now, including the correlation between the SSRI and SNRI classes. We agree with the reviewer that this is a complete approach. Our results remained essentially unchanged. Some results became more sensible, like the consistency of effects for nausea and vomiting and suicide thoughts and suicide attempt. We have now substituted the figure and the results in Supplementary Table S2 and the description of the results.

4. On page 8, there are few statistics reported regarding the SEM models in the 'Side effects co-occur across medications' section. Many statements were not supported by statistical evidence. For example, 1) in 'we identified high correlations for side effects across medications', a range of correlation coefficients should be given;

We have now added the median correlations in the text and a new supplementary figure to support this sentence, which now reads:

"We identified high correlations for the same side effects across medications (within side effect median correlation=0.57, within medication median correlation= across side effect median correlation= 0.27 see Supplementary Figure 2)."

Supplementary Figure 2 contains the following new figure:

Supplementary Figure 2 Side effect correlation distribution across medications

Density plot depicting the kernel density estimates based on the distribution of side effects correlations shown in Figure 2. a) Data split between the correlations for the same side effects across medications (blue) or different side effects within and across medications red. b) Data split between the correlations for different side effects within the same medication or across medications

2) in 'A clustering based on these correlations grouped these variables by side effect rather than by antidepressant or medication class', model statistics should be given to support the conclusion for model comparison.

We believe the figure above further supports this trend. However, we have rephrased this sentence to express that we refer to the visual trend of the clustering, given that model fitting follows this result.

5. In addition to point 4, only AICs were reported in Supplementary Table 2 for SEM models. I encourage the authors to include other measures such as CFI, TLI, RMSEA.

We have now included the requested indices in the supplementary table. We note that the best way to compare the proposed models is to use relative indices such as AIC and likelihood ratio tests. Still, we agree with the reviewer that these measures are important for evaluating overall model fit.

Similarly, it would be necessary to report association strengths in Supplementary Figure 2 or at least a range of effect sizes for parallel tests.

Supplementary Figure 2 (now supplementary figure 3) depicts the tested models. Each of the 25 side effects was explored using these models (across the ten medications). There would be 25 values for each of the loadings in the pictures. There is no reason for the loadings to be consistent across side effects, as some side effects fit better different models (i.e. those with an SNRI factor). Because of this, we believe that adding all of the effects to the diagram would be impossible and could result in confusion if summarized within the figure.

Reviewer #2 (Remarks to the Author):

The paper is an investigation into the effects of genetic loading for depression, BMI, insomnia and headaches on self-reported anti-depressant side effects. They used an Australian depression cohort that had details on participant's genetics, class of antidepressants taken, as well as a list of 23 side effects. The paper makes use of both structural equation modelling and sophisticated PRS generating software (SBayesR) which is a major plus point of the manuscript. The paper concludes that antidepressant side effects could well be driven not just by class of antidepressant but in combination with an individual's genetic loading for the tested traits.

I think that this paper is a useful addition to the current literature in a very important area of research. The paper is concise, written with a sufficient level of detail for other to reproduce the analyses and the conclusions are well thought out. However, I do not think the paper is suitable for publication in its current form without 1 major change to the methodology. For BMI and chronic headaches, summary statistics from Neale's public repository of UK Biobank GWAS statistics were used. The use of these outputs can be useful but as they are generated using an automated pipeline they have minimum data cleaning, and as such, should not be used as a basis for a publication if other summary statistics are available. The analysis should be reran using more appropriate summary statistics for BMI and chronic headaches. Here are two potential papers that would be more appropriate.

Yengo et al, BMI GWAS/meta-analysis

<https://www.ncbi.nlm.nih.gov/pmc/articles/PMC6488973/>

Gormley et al, migraine GWAS

<https://www.ncbi.nlm.nih.gov/pmc/articles/PMC5331903/>

We thank the reviewer for the comments and constructive feedback. We used BMI PRS from the Neale GWAS, but we performed the headaches GWAS and described it in a recent publication exploring causal associations with migraine (10.1186/s10194-021-01284-w). We forgot to add the reference as the paper was still in press. Following the suggestion, we have now performed the analyses using PRS derived from Yengo et al. The association between BMI PRS and weight gains became stronger. The summary statistics for migraine are not publicly available (only SNPs with $p < 10^{-5}$ are available). Because of this, we could not pursue the second part of this comment, but we do note that chronic headaches showed a substantial overlap ($r_g = 0.85$) with the 23andMe migraine GWAS as reported in Garcia-Marin *et al.* (see 10.1186/s10194-021-01284-w).

A two minor comments:

- Although 23 side effects were investigated there were only 5 risk scores created. The paper should go into more detail as to why those particular traits were selected to create risk scores and not at genetic loading of any of the other side effects.

We focused on traits for which we identified readily available sufficiently powered GWAS of traits that are highly related to the side effects under study. For example, a headaches or migraine PRS would be reasonably close to headaches as a side effect. BMI is also a natural link to weight gains and insomnia to trouble sleeping. For other side effects (e.g. reduced sexual function, nausea, dry mouth, runny nose, etc.)

there are no GWAS available (to the best of our knowledge). Furthermore, other traits such as anxiety or suicide attempt do have GWAS, but are still underpowered. We have modified the sentence in page 9 to read:

"We chose to study weight gain, insomnia and headaches as we could identify related complex traits (BMI, insomnia, and headaches) for which well-powered GWAS data is readily available."

- Figure one has lines connecting the data points, but the x axis scale is categorical. The plots could be made clearer by removing the line connecting the data points and using box plots instead with males and females side by side for each category.

The reviewer raises a good point. However, we cannot use boxplots or other distribution visualizations because these are prevalence point estimates. We tried removing the lines (see below), but believe the result makes the estimates harder to compare:

Thus, we have opted for keeping these figures as they are. In our opinion, the lines (and the slopes in particular) aid in identifying the differences. Nonetheless, we are willing to search for other representations if the reviewer or the editor feel strongly about it.

Reviewer #3 (Remarks to the Author):

This is an interesting paper, however it is not clear what is really new, what is the take home message (about the relationship between individual antidepressants and specific adverse events) and how the authors validated their findings (see limitations in the discussion section).

We thank the reviewer for their time assessing our manuscript. Our study has several novel findings including: 1) assessing the prevalence of reported side effects outside of a clinical-trial. 2) Test for drug-class specificity of antidepressant side effects. These results are evident from the prevalence estimation, the correlation and the structural equation models performed; 3) testing and providing evidence in favor of non-specific factors (i.e. not related to the type of medication) and 4) further testing for the genetic basis of these general factors using genetic instruments. These are insights that remained to be understood and tested. We have now explicitly mention this in the first paragraph of the discussion

There sample size of the controls is relatively very small; moreover, the retrospective design of this study and the recollection bias (as reported by the authors themselves) are important concerns that limit the reliability of these findings.

We have disclosed the retrospective design of this study in the limitations section. We do not think retrospective studies are without value. It is unclear to us to what the reviewer means by controls, but this is the largest independent investigation on the aetiology of antidepressant side effects.

Many of the references are old or not the most appropriate/relevant.

We could not address this comment given the lack of detail (what references are irrelevant, which other manuscripts would the reviewer consider to have been missed).

I think this article would benefit from the replication of the findings in a different cohort and from the inclusion of clinical researchers with expertise in antidepressant studies among the co-authors.

We agree that replication in a different cohort is a gold standard and we strive to achieve it. However, there is only another study, which is still undergoing recruitment and genotyping, that has sufficient phenotyping depth to replicate our results. Replication will be sought in the future when genotyping is finished. Among the co-authors, there is a psychiatrist with years of experience in both the clinic and research. The study has certainly benefited from their expertise.

There is no protocol, so it is not clear why and how the authors selected the list of side effects.

The protocol for the AGDS was published recently (10.1136/ bmjopen-2019-032580). Further we explicitly stated: "*For each antidepressant taken, participants were asked whether they had experienced side effects and, if they did, to select which from a checklist with **the twenty-three most commonly reported antidepressant side effects including***"

Tables and figures should be more informative

We do not agree. Our tables are informative. Reviewer two had a minor comment about figure 1 and we have explored the possibility of changing it (see above), but decided against doing so because the result was harder to interpret.

REVIEWERS' COMMENTS:

Reviewer #1 (Remarks to the Author):

The authors addressed my comments thoroughly and sufficiently. I have no additional comments.

Reviewer #2 (Remarks to the Author):

I feel that the reponses to the reviwer comments are satisfactory and a such feel the manuscript is suitable for publication.